# Determinants of anxiety levels among young males in a threat of experiencing military conflict–Applying a machine-learning algorithm in a psychosociological study

Iuliia Pavlova[1], Dmytro Zikrach[2,3], Dariusz Mosler[4]*, Dorota Ortenburger[4], Tomasz Góra[4], Jacek Wąsik[4]

1 Theory and Methods of Physical Culture Department, Lviv State University of Physical Culture, Lviv, Ukraine, 2 SoftServe, Lviv, Ukraine, 3 AiNanoLab, Dubai, United Arab Emirates, 4 Department of Health Sciences, Jan Dlugosz University in Czestochowa, Częstochowa, Poland

* dariusz.mosler@gmail.com

## Abstract

### Background

Anxiety could be felt even in objectively peaceful situations, but a vision of conflict could result in increased stress levels. In this article, we aimed to identify hidden patterns of mental conditions and create male profiles to illustrate the different subgroups as well as determinants of anxiety levels among them in accordance with proximity to a possibility of direct exposure to military action.

### Methods

A sample of Ukrainian males, in duty as conscripts to military service (n = 392, M±SD = 22.1±5.3) participated in a survey. We used the 36-Item Short Form Health Survey, and State-Trait Anxiety Inventory. In addition to psychological indices, social-demographic data were collected. To discover the number of clusters, the k-means algorithm was used, the optimal number of clusters was found by the elbow algorithm. For validation of the model and its use for further prediction, the random forest machine-learning algorithm, was used.

### Results

By performing k-means cluster analyses, 3 subgroups were identified. High values of psychological indices dominated in Subgroup 2, while lowest values dominated in Subgroup 3. Subgroup 1 showed a more even distribution among the indices. The strength of the relevance and main determinants of the prediction of the presented model mostly consisted of mental qualities, while socio-demographic data were slightly significant.

**Data Availability Statement:** All relevant data are available on GitHub (https://github.com/zikrach/quality_life_research).

**Funding:** The author(s) received no specific funding for this work.

**Competing interests:** The authors have declared that no competing interests exist.

## Conclusions

There is no clear relevance between proximity or even the experience of military actions and anxiety levels. Other factors, mostly subjective feelings about mental conditions, are crucial determinants of feeling anxiety.

## Introduction

Anxiety is an emotion that could be felt in certain situations, or in some cases, it could be a prolonged state that leads to other disturbances or even disorders. It is characterized by feelings of tension, worriness, and constant readiness for fight or flight responses. It could be felt during stressful situations that involve the necessity of confronting some problems in socially stressful situations [1]. This type of behaviour usually does not occur during physical confrontations, when fight or fight responses are involved. However, experiencing such conflict may lead to fear of another one and prolonged readiness for such fight [2]. Another option is when social (verbal or situational) conflict took place. In this type of action, there was no physical contact with the opponent, but stress hormone levels were elevated anyway [3]. Moreover, in such situations, the end of the interaction did not result in the end of the physiological response. When such phenomena occur, a prolonged version of anxiety will start to emerge [4]. This state of being constantly ready for conflict that may in fact never happen is sometimes defined as feeling that "something is going to happen" [5]. This thought can lead to overall tension in the body, muscle stiffness, increased blood pressure and other physiological consequences connected to increased cortisol and amylase levels [6]. Being under stress may be expressed by physiological symptoms such as stress, trembling, dizziness, and rapid heartbeat [7]. The longer individuals are in this state, the worse the consequences for cognitive abilities are. A lack of concentration, an inability to memorize things and even having the impression of hearing high frequency sounds all the time can occur. This is due to the impact of cortisol in the amygdala and hippocampus [8]. The mechanisms described above are the same in every stressful situation. The consequences are also the same–fear of yet-to-come conflict and a state of preparedness. With the exception of peaceful zones, the same stress-inducing mechanisms are present among people who live near armed conflict zones or fight in combat zones [9]. Typically, all anxiety-related changes were believed to be connected to post-traumatic stress disorder (PTSD). Historically, this syndrome was first described after World War I, and most extensively after World War II. This term was defined based on observation of veterans and wounded soldiers when they returned from the front [10]. However, these cases involved trauma, physical injury and direct or indirect hits; a direct hit is being shot, wounded by shrapnel or experiencing a shockwave from an explosion, whereas an indirect hit is being close enough to feel the pressure of being under fire without experiencing any physical injuries [11]. Currently, PTSD is no longer associate with the direct physical stressor, but also is believed to be caused by indirect exposure to aversive details of the trauma or learning that a relative or close friend was exposed to a trauma [12].

People who are preparing for combat and civilians who live close to conflict may also feel anxiety because of such events [13]. Currently, the media can transmit information from battlefields, so not only people who directly hear gunshots or artillery but also the whole country or province can become anxious, which occurs with certain events [14]. Assuming that there are differences in the factors that induce the stress, the anxiety responses of the body are the

same, and therefore prolonged exposure to these feelings could be comparable in different situations, such as states of war and peace.

Anxiety is an emotion that is studied by both civilian and military psychologists. To have comparable diagnostic mechanisms, the State-Trait Anxiety Inventory (STAI) was developed [15]. Through many years and numerous studies, the STAI has proven to be a reliable tool for assessing the diagnosis and degree of anxiety among many groups of people with different ages, genders, professions, etc. Research with the STAI has contributed to understanding that there are differences between the degree of anxiety (its intensity) as a function of individual differences between people and the degree of anxiety as a function of being prone to the development of stress as a personal trait.

Regarding extreme environmental and social conditions, such as war and armed conflicts, these two types of anxiety should be concurrently elevated. Answers in the state anxiety part of this inventory indicate changes in the anxiety level connected to changes in ongoing events and the stress induced by them. The trait anxiety part covers personal traits, which allow the differential diagnosis of individual capabilities of handling stress. Assuming stress levels due to ongoing events as a constant, different results on the STAI inventory will be due to individual traits of a person. Following this reasoning, individual traits are also connected with cultural and behavioural conditioning, which are dependent on the environment in which one grows up, such as the country where individuals were born. In regions on the edge of combat zones, these traits will determine the anxiety level of the population where the conflict occurs. The STAI has been used for over 50 years with many archival results that allow comparisons between experimental groups from new research with older groups, increasing its credibility.

The STAI was used to examining trait and state anxiety in different military population [16–19], and this tool is a recommended method approved by the Supreme Council of Ukraine for diagnostic examination and psychological correction of military personal, especially those individuals who took a direct part in the anti-terrorist operation. However, despite the availability of legal documents, described recommendations and procedures, any analysis or cohort studies of anxiety and depression was conducted on large population groups of Ukrainian military personal, there is no clear data on the prevalence of PTSD among military personnel.

The collected databases do not include a population that faces modern conflict such as the conditions in Ukraine that have been ongoing since 2014. The conflict started with the increased insurgency against the newly established government after the removal of Yanukovych government in the south-east region of Ukraine. Then armed conflict started as Russian soldiers without any military insignia manipulated those areas and declare separation from the rest of the country and finally annexation of Crimea, which is not recognized by international institutions. Since then, there was an anti-terrorist operation ongoing, which is officially held by the Ukrainian military since 2018, when president Poroshenko announced that aggression is caused by the Russian Federation. Conflict status is ongoing. Ukraine has conscript army, so the students of different military-related studies will have to be on duty at some point.

Another useful tool is 36-item Short Form Health Survey, which was used for assessing health-related quality of life outcomes, and it is validated for general and military population in different countries [20–22]. The questionnaire is widely used to assess the quality of life of the Ukrainian population aged 16–70, in particular for people working in emergency services [23].

Environmental factors and anxiety-related consequences strongly affect overall quality of life [24]. It would be only a partial diagnosis if one survey was completed without a context of the overall perception of quality of life for the people who are assessed.

In the era of big data and the development of machine learning algorithms, multifactorial analysis in sociology has become easier [25]. Previously, analyses were limited to a certain

number of factors, or their presentation required enormous graphs and computations to obtain a coherent vision of important factors to obtain a synthetic vision of crucial factors regarding some phenomena. In recent years, computational social science has attained a place among methods of analysis for psychosociological data [26]. Standard survey-based measures of happiness could then be used to train prediction models with use of computation models from "big data" sources, which allowed for a finer analysis across time and space of the determinants of well-being. The usefulness of these analyses was well expressed by a WHO report stating that in the era of big data and the development of machine learning algorithms, multifactorial analysis in sociology became easier [27].

The idea of using machine-learning algorithms to obtain knowledge about PTSD and anxiety of soldiers is not novel. There were even large cohort studies, including 13 690 participants, in which a supervised machine-learning algorithm was applied [28]. The main results of such an experiment conducted by Leightley et al. indicated, that such methods might reduce cost, and helps with earlier detection and prophylaxis [28]. Another approach, with feature selection and k-nearest neighbors, was used by Karstof et al., with further confirms the possibility to use a machine learning as a forecast of PTSD symptoms on a group of 561 Danish soldiers [29]. Another machine learning classifier random forest, for soldiers that participated in operation in Afghanistan [30]. Also, in this case, the sample cannot be considered as big data (n = 473). Despite that, machine learning algorithms have proven to be worthy of developing and capable of adjusting to a different group of soldiers participating in various military conflicts.

The main assumption is that the closer a person is to a conflict zone or that the vision of participation in a fight becomes more real, the higher the level of anxiety and the lower the quality of life. In this study, it is understood as different kind of military-related studies brings different chances for actual participation in direct military action for students.

Therefore, the aim of this study was to obtain knowledge about the relationships between risk for participation in direct armed conflict and the level of anxiety. Further purpose of this study was to assess of the quality of life for people from different groups of military related students.

In this article, we aimed to explore the heterogeneity in a young male population in excellent physical condition, identify hidden patterns of mental conditions, and create male profiles to illustrate the different subgroups found within the complex population, as well as to identify determinants of anxiety levels among them in accordance with proximity to the possibility of direct exposure to military action.

## Method

The framework was composed of several phases: [1] data selection; [2] data preprocessing; [3] determination of the number of clusters and their interpretation; [4] learning by classifier, validation of obtained model, and achieving a trained model with highest accuracy; and [5] using the trained model for the prediction of the testing part of the data (20%).

The questionnaire for data collection consisted of the following: sociodemographic data, information about military service, and questions for the assessment of quality of life (36-Item Short Form Health Survey, Ukrainian version) and level of anxiety (State-Trait Anxiety Inventory). This study was approved by Bioethics Committe of Lviv State University of Physical Culture.

### Participants

A sample of Ukrainian males (n = 392, M±SD = 22.1±5.3) participated in a survey.

Four groups of respondents were involved in the study:

- students of military specialization (n = 123, M±SD = 21.2±5.8),

- internal affairs specialization (n = 101, M±SD = 19.1±0.5),

- sport specializations (n = 64, M±SD = 19.7±1.0),

- military service personnel (n = 104, M±SD = 25.3±4.7).

The distribution and appointment of conscripts were carried out in proportion to the need and availability of conscription resources, both for the Armed Forces of Ukraine and other military formations. Young people were distributed, taking into account their moral, business, psychological qualities, state of health, physical development, general education, and specialized training, as well as taking into account the need for military reserves. Students from those profiles were most capable for military service, because, in accordance with the curriculum, they were given military-related training such as firearms or proper physical preparation. Although all students in Ukraine had a mandatory conscription process, the directives said that after a minimal amount of preparation time, students from those groups were fastest in terms of being sufficiently trained.

Universities offers reserve officer training for two years. An alternative to that is one-year regular service as conscript troops. Maximum conscription age is 27 years.

Students in our sample were conscripted for ground type troops such as mechanized infantry, artillery and engineering troops. Total amount of Ukrainian ground forces are 145 000 soldiers. According to good statistical practice, Cochran's Sample Size Formula was applied. With 5% of margin of error and confidence level of 95%, required sample size was 375 participants, therefore our sample was representative [31].

Personal characteristics of participants included gender, age in years, education level (secondary education, bachelor's degree, master's degree, Ph.D. degree), marital status (never married, married, divorced/live separate, widow/widower, civil marriage), length of service (in months), residence time in conflict zone, the existence of war injuries, and duration of rehabilitation (in months) if person sustained injury in the military zone. Sociodemographic data reflected participants' status as of September, 2018.

All participants agreed to engage in the study and provided written informed consent. The participants were informed about the purpose of the study and were informed that their participation would be voluntary and that they could withdraw from the study at any moment. The participants were also apprised that the collected information would be anonymous, and individual data could not be identified.

After giving written informed consent, each participant was asked to complete the self-administered questionnaire in hard copy.

### 36-Item Short Form Health Survey (SF-36)

The Ukrainian version of the SF-36 consists of 36 questions grouped into 8 scales: physical functioning (PF), physical role functioning (RF), bodily pain (BP), vitality (VT), social functioning (SF), mental health (MH), emotional role functioning (RE), and general health (GH). Questions provided useful information on the health status of the respondents, namely, a subjective evaluation of physical activity level, the impact of physical and emotional health on daily activities, limitations due to pain, the presence of low energy and tiredness, the quality of respondent's social relations, difficulties connected with mental states, social communications, expectations of changes in health in the future, etc [9].

A score ranging from 0 (indicating the worse quality of life) to 100 (indicating the best quality of life) was assigned for each scale. Scale scores were summarized into a Physical

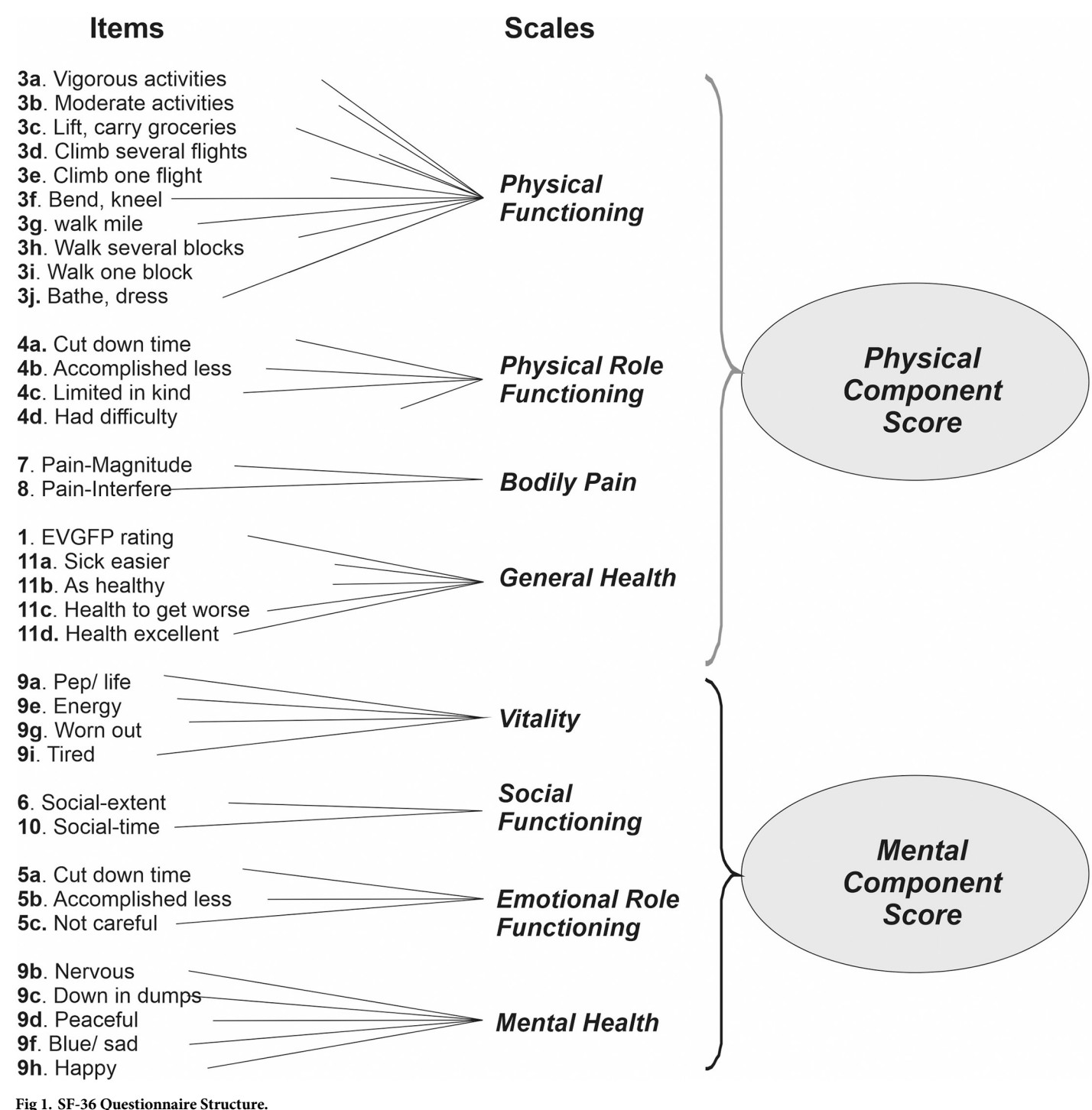

**Fig 1. SF-36 Questionnaire Structure.**

Component Score (PCS) and Mental Component Score (MCS). The PCS contained physical functioning, physical role functioning, bodily pain and general health; the MCS consisted of vitality, social activity, mental health, and emotional role functioning (Fig 1). For the PCS and MCS, a score below 50 indicated a worse quality of life compared to the average indices in the general population.

**State-Trait Anxiety Inventory.** We used a psychological self-report inventory to assess the presence and severity of current symptoms of anxiety and a generalized propensity to be anxious. The STAI has 40 items, with 20 items allocated to each of the S-Anxiety and T-Anxiety subscales (4-point Likert scale). Two types of anxiety are determined based on the measurements within the STAI. First, the State Anxiety Scale (S-Anxiety) evaluates the current state of anxiety, asking how people feel "right now" and uses items that measure subjective feelings of apprehension, tension, nervousness, worry, and activation of the autonomic nervous system. Second, the Trait Anxiety Scale (T-Anxiety) is connected with predispositions (personal traits) and perceived anxiety. Responses to items on the S-Anxiety scale (to assess intensity of current feelings "at this moment") were as follows: 1) not at all, 2) somewhat, 3) moderate, and 4) very much so. Responses to items on the T-Anxiety scale (to assess the frequency of feelings "in general") were as follows: 1) almost never, 2) sometimes, 3) often, and 4) almost always [10].

For the purpose of additional insight into the result, we also used transformed raw scores of the test into stens scores. It was possible due to previous studies with normalization of such results on a group of Polish soldiers from regular conscription, which can be considered as a corresponding population to the groups tested in this study. That study served as a frame of reference for the transformation of our result to 10 points stens score, which represents ten sections with a range of raw results points. Normalization studies were conducted by Laboratory of Psychology Test of Polish Psychological Society [11].

**Data preprocessing.** When analysing the obtained data, missingness was analysed, and data of participant with missing values (any quality of life, anxiety features, sociodemographic characteristics, etc.) were dropped. Finally, the data of 278 participants were used for further analyses. Binarization was used to transform categorical data features (education level, marital status, place of study/work) into vectors of binary numbers to make classifier algorithms more efficient [12].

All quality of life features before clustering were categorized as low (0–50 points), medium (50–75) and high (75–100 points) and were also converted to binary variables. Continuous data were expressed as the mean (M), median value (Me), standard deviation (SD), standard error (SE), variance ($\sigma^2$), confidence intervals (95% Cl), minimum (Min), and maximum (Max).

**Determination of the number of clusters and their interpretation.** To reveal hidden data structures in the data set, cluster analysis was used. The variables used for clustering were quality of life and anxiety indices (PF, RF, BP, VT, GH, SF, RE, MH, T-anxiety, and S-anxiety). For visualization, we applied the 2D projection of feature space by the t-SNE algorithm.

To discover the number of clusters, the k-means algorithm was used; this is a data partition method used in many fields, including data mining, pattern recognition, and decision support. K-means clustering is a way to use data to uncover natural groupings within a heterogeneous population. The aim of this algorithm is to find groups in the data, with the number of groups represented by the variable $k$. The algorithm works iteratively to designate each data point to one of $k$ groups based on the features that are provided. Centroid-based models represent clusters by a central vector, which does not need to be an actual object. To identify patterns, the algorithm starts by first assigning data points into random groups, then the group centres are calculated and the group memberships are reassigned based on the distances between each data point and the group centres. The optimal number of clusters was found by the elbow algorithm, which is based on the sum of the squared distances between data points and their assigned cluster centroids. This method is used to identify the maximal number of efficient clusters. Adding more clusters than obtained from an elbow algorithm does not contribute to better modelling of data.

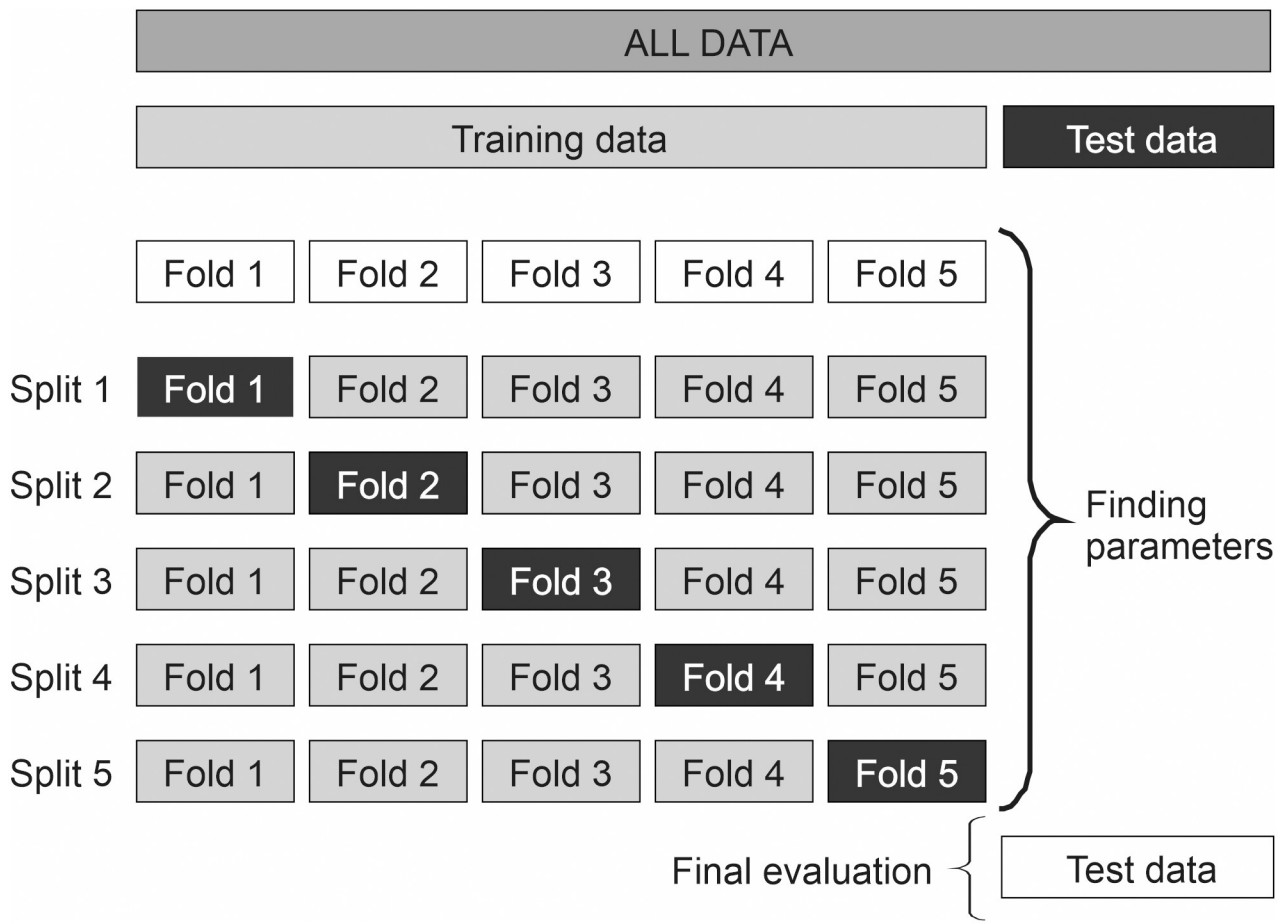

**Fig 2. Visualization of algorithm adjusting data.**

Variables for describing obtained clusters included age, marital status, length of military service, residence time in conflict zone, the existence of war injuries, and duration of rehabilitation.

**Validation of the model and its use for further prediction.** A classifier, the random forest machine-learning algorithm, was used. It applies the technique of bootstrap aggregating to decision tree learners and uses averaging to improve predictive accuracy and control over-fitting. The performance of the model was tested by the cross validation technique (Fig 2). Data was split randomly with shuffling and stratifying based on the target into groups for training the model purposes (80%) and some portion of the data (20%) was kept aside for a purpose of testing obtained models. After training the model, these test data was used for validation. The training set was randomly split into 5 smaller sets with stratifying based on the target, and the model was trained using four of the folds as training data. The obtained model was validated on the remaining fold, and the test data were used to compute the accuracy of the model.

At these stages, a variety of selected data were used (sociodemographic characteristics, quality of life, anxiety results in numeric and categorical form). Quality of life scores according to different scales were used for anxiety prediction.

**Table 1. General characteristics of socio-demographic data.**

| Items | M±SD (95% CI) (Min; Max) | | | |
|---|---|---|---|---|
| | Students of military specialization | Students of internal affairs | Students of sport specialization | Military service personnel |
| Age, years | 21.2±5.8 (20.3–22.2) (16; 48) | 19.1±0.5 (19.0–19.2) (18; 22) | 19.7±1.0 (19.4–19.9) (19; 25) | 25.3±4.7 (24.4–26.2) (20; 43) |
| Duration of military service, month | 36.0±52.9 (27.7–44.3) (12; 276) | 2.4±9.8 (0.5–4.3) (0; 48) | 0.0 | 55.0±51.2 (45.0–64.9) (0; 288) |
| Residence stay in conflict zone, month | 2.4±5.7 (1.5–3.3) (0; 38) | 0.0 | 0.0 | 10.2±10.7 (8.1–12.3) (0; 71) |

## Results

The characteristics of the total cohort are presented in Table 1. The participants were students of military (n = 158, age: M±SD = 21.2±5.8 years, duration of military service: 0–276 months, residence stay in conflict zone: 0–38 months), internal affairs (n = 101, age: 19.1±0.5 years) and sport (n = 62, age: 19.7±0.1 years) specializations, and military service personnel (n = 104, age: 25.3±4.7, duration of military service: 1–288 months, residence stay in conflict zone: 0–71 months).

We estimated the quality of life for each scale in the SF-36 for every social group (Table 2). High homogeneity in the population was identified, and the $\chi^2$ test results were nonsignificant (p > 0.01) for all quality of life scales between subgroups.

The calculation of SF-36 indices showed that the lowest mean scores for quality of life were related to bodily pain, general health, vitality, and mental health scales, and the highest scores were related to the physical functioning scale. The variance indicated that all values across scales were not identical, and the numbers in the sets were far from the mean.

For STAI results, the highest mean of T-anxiety was observed in a group of students of sport specialization (M = 45.5), and the lowest in the group of military service personnel

**Table 2. Subgroup analysis of quality of life (M±SD; 95% CI; σ²).**

| Scales | Students of military specialization | Students of internal affair specialization | Students of sport specialization | Military service personnel |
|---|---|---|---|---|
| Physical functioning | 89.8±22.3; (86.1–93.5); 497.5 | 95.2±8.2; (93.5–96.8); 67.1 | 93.6±13.7; (90.1–97.0); 187.6 | 91.1±16.4; (87.8–94.5); 268.0 |
| Physical role functioning | 73.8±32.4; (68.5–79.1); 1047.4 | 75.3±27.7; (69.6–80.9); 764.6 | 76.7±25.6; (70.1–83.3); 654.0 | 79.4±31.3; (73.1–85.7); 976.7 |
| Bodily pain | 68.8±24.3; (64.8–72.7); 589.8 | 70.7±22.7; (66.2–75.2); 515.6 | 67.8±22.2; (62.2–73.5); 491.1 | 71.7±23.5; (67.1–76.4); 553.1 |
| General health | 66.1±20.1; (62.8–69.4); 404.0 | 68.7±16.5; (65.4–71.9); 271.0 | 60.9±15.4; (57.0–64.9); 238.1 | 71.6±17.1; (68.2–75.0); 291.7 |
| Vitality | 67.6±16.3; (65.0–70.3); 264.7 | 64.7±19.7; (60.8–68.6); 387.9 | 65.0±15.2; (61.1–68.9); 230.0 | 68.9±19.3; (65.1–72.7); 371.5 |
| Social Functioning | 76.2±25.8; (72.0–80.3); 667.7 | 81.3±19.8; (77.4–85.3); 393.4 | 82.1±21.1; (76.7–87.4); 443.8 | 86.5±18.3; (82.8–90.1); 333.8 |
| Emotional role functioning | 76.3±34.7; (70.7–81.9); 1205.6 | 78.4±34.4; (71.4–85.3); 1181.5 | 77.6±33.8; (68.7–86.5); 1145.7 | 83.2±27.9; (77.5–88.8); 777.2 |
| Mental health | 67.1±18.9; (63.9–70.3); 358.3 | 68.5±18.1; (64.9–72.2); 328.5 | 71.0±14.8; (67.1–74.9); 218.7 | 73.1±17.6; (69.5–76.6); 308.0 |
| Physical component score | 50.8±6.0; (49.7–51.9); 36.2 | 52.2±4.9; (51.2–53.3); 24.3 | 52.2±6.2; (50.5–53.9); 38.4 | 50.7±5.0; (49.6–51.9); 25.1 |
| Mental component score | 47.7±9.7; (45.9–49.5); 94.4 | 47.5±9.6; (45.4–49.6); 91.4 | 48.4±9.0; (45.9–50.9); 80.7 | 51.2±7.7; (49.4–52.9); 58.7 |

**Table 3. Results of the STAI questionnaire for specific groups of participants.**

| | n | M | SD | SE | 95% CI | $\sigma^2$ | Min | Me | Max |
|---|---|---|---|---|---|---|---|---|---|
| *Students of military specialization* | | | | | | | | | |
| T-Anxiety | 129 | 41.4 | 8.7 | 0.8 | 39.9–42.9 | 76.0 | 23.0 | 42.0 | 70.0 |
| S-Anxiety | 139 | 40.6 | 8.8 | 0.8 | 39.1–42.1 | 78.2 | 21.0 | 40.0 | 64.0 |
| *Students of internal affair specialization* | | | | | | | | | |
| T-Anxiety | 93 | 43.9 | 7.6 | 0.8 | 42.4–45.5 | 57.5 | 29.0 | 43.0 | 62.0 |
| S-Anxiety | 92 | 42.0 | 7.1 | 0.7 | 40.5–43.5 | 49.9 | 23.0 | 42.0 | 66.0 |
| *Students of sport specialization* | | | | | | | | | |
| T-Anxiety | 57 | 45.5 | 8.9 | 1.2 | 43.2–47.9 | 78.4 | 29.0 | 45.0 | 69.0 |
| S-Anxiety | 62 | 40.8 | 7.9 | 1.0 | 38.8–42.8 | 62.6 | 23.0 | 41.0 | 59.0 |
| *Military service personnel* | | | | | | | | | |
| T-Anxiety | 95 | 40.0 | 7.5 | 0.8 | 38.5–41.6 | 56.5 | 23.0 | 41.0 | 54.0 |
| S-Anxiety | 96 | 39.0 | 8.2 | 0.8 | 37.3–40.7 | 67.8 | 20.0 | 40.0 | 60.0 |
| *War veterans with injuries* | | | | | | | | | |
| T-Anxiety | 19 | 43.4 | 9.4 | 2.1 | 38.9–47.9 | 87.7 | 24.0 | 45.0 | 54.0 |
| S-Anxiety | 20 | 43.7 | 9.8 | 2.2 | 39.1–48.3 | 95.2 | 23.0 | 44.5 | 60.0 |

(M = 40). In this group also the lowest and the highest value of T-Anxiety indicator for certain individuals were observed.

The highest mean of S-Anxiety results was observed in war veterans with injuries (M = 43.7), while the lowest mean was indicated in the group of military service personnel (M = 39). The lowest value was observed in a group of military service personnel (Min = 20), while, the highest in the group of internal affair specialization (Max = 66) (Table 3).

Analysis of the obtained results, especially those transformed into the stens scale, showed the highest differences between stens of both anxiety indicators in a group of students of military specialization. The lowest obtained values correspond to the 1st sten, and the highest values correspond to the 9th sten (in the case of T-anxiety Trait, symptom frequency). The results show that the range was from extremely low to extremely high values, as every group spread of results for at least 7 points on 10-point scale (Table 4).

## Cluster analysis

By performing k-means cluster analyses, 3 clusters (subgroups) were identified, with cluster sizes ranging from 45 to 120 participants (Fig 3).

**Table 4. Results of the STAI standardized scores (1–10).**

| | T-Anxiety Min Trait—symptom frequency | T-Anxiety Max Trait—symptom frequency | S-Anxiety Min State–symptom intensity | S-Anxiety Max State–symptom intensity |
|---|---|---|---|---|
| | *Standardized Scores (1–10)* | *Standardized Scores (1–10)* | *Standardized Scores (1–10)* | *Standardized Scores (1–10)* |
| Students of military specialization | 1 | 9 | Below 1st sten | 8 |
| Students of internal affair specialization | 2 | 8 | Below 1st sten | 8 |
| Students of sport specialization | 2 | 9 | 1 | 7 |
| Military service personnel | 1 | 7 | Below 1st sten | 7 |
| War veterans with injuries | 1 | 7 | 1 | 7 |

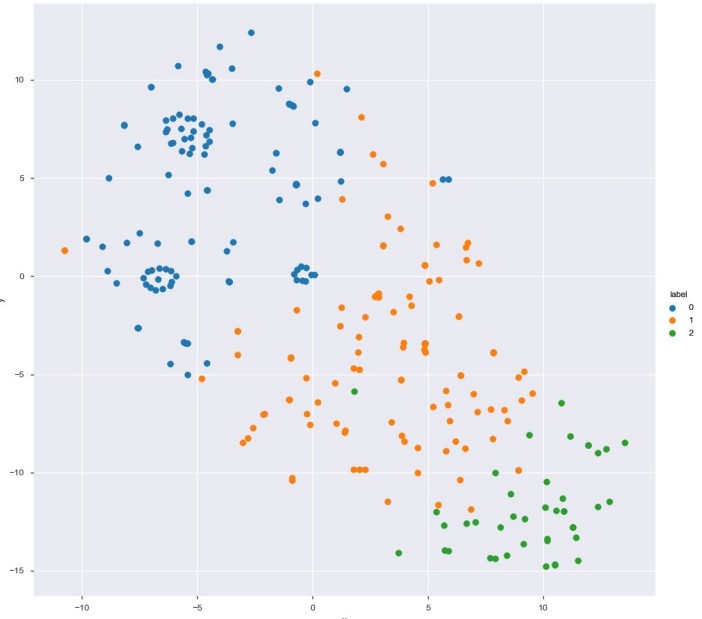

**Fig 3. Visualization of cluster analysis results.**

For better visualization of the distribution of participants in each cluster from different groups of specialization (Fig 4) and psychological indices presented by them (Figs 5 and 6), we also present data in the form of graphs. For all three figures, the proportions of indices were similar, which corresponded with the mechanics of the algorithm, which is design to find similarities between variables and group them. High values dominated in Subgroup 2, and the

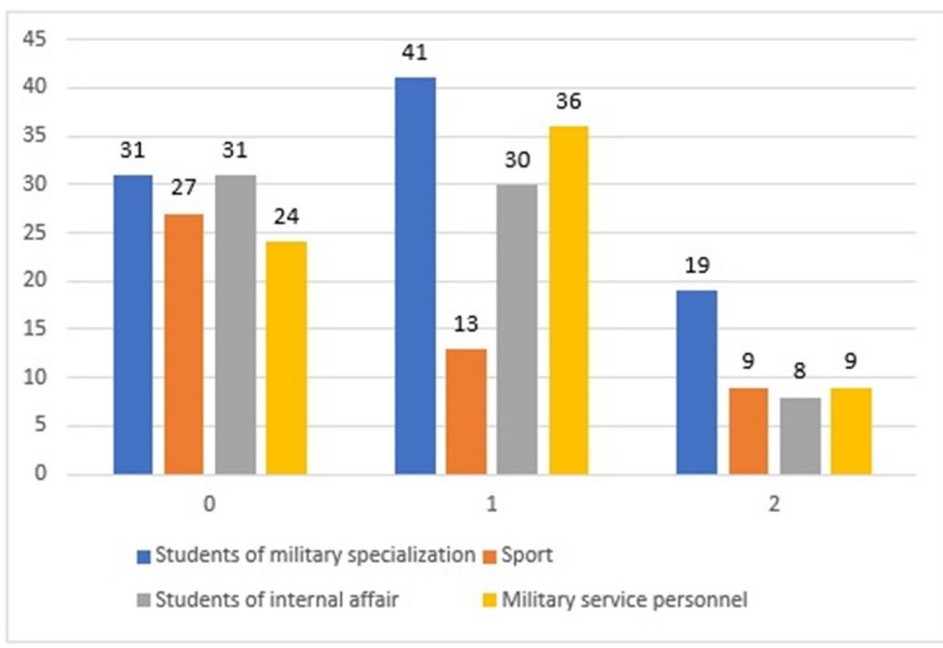

**Fig 4. Distribution of amount of participants from different groups in each cluster.**

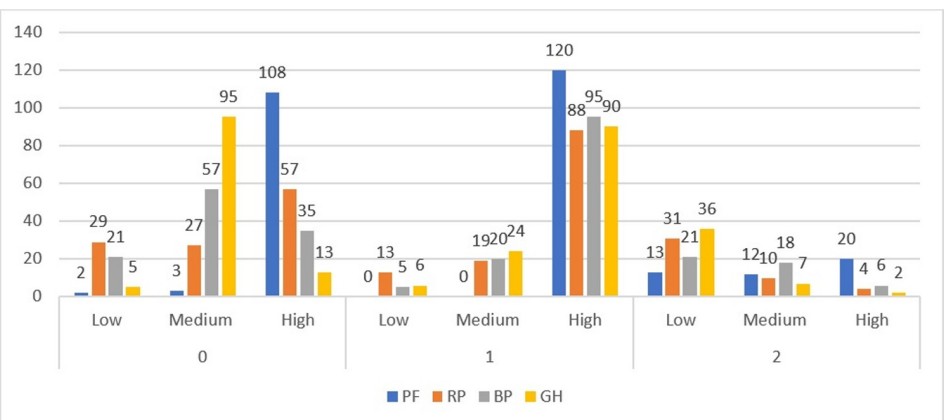

**Fig 5. Distribution of amount of participants by physical health indices in each cluster.**

lowest dominated in Subgroup 3. Subgroup 1 showed a more equal distribution of the indices in each level.

The cluster analysis allocated individuals into subgroups based on the characteristics they have in common. The basic characteristics of these clusters are shown in Tables 5 and 6.

The subgroups were characterized by high age homogeneity, whereas Subgroup 3 connected the persons with a high duration of military service, experience of war injuries and/or experience of rehabilitation (Table 5). This cluster included married (official, civil) participants (17.78%), whereas in clusters 0 and 1, the related numbers were 9.73% and 4.16%, respectively. Subgroup 3 consisted mainly of students of military specialization (42.22%), whereas cluster 1 had the highest percentage of persons from military services (Table 6).

Subgroup 1 (n = 113), the second largest cluster, consisted of young males with a wide range of places of study/work. The proportion of students from universities with strike discipline (military, internal affair) was 54.86%, and the number of single persons was 87.61%. Quality of life indices according to all scales where the second highest, and cluster 0 showed the presence of participants with high physical functioning (95.58%), self-functioning (61.95%), and role-emotional (62.83%) scores. Crucial for a significant proportion of participants in this cluster was the low level of role-functioning (25.66%) and vitality (20.35%). Subgroup 1 showed the highest weight of participants (50.44–84.07%) with medium quality of life indices according to the quality of life scales that characterized both physical and mental health. Quality of life within almost all scales (bodily pain, general health, vitality, and mental health) was in the medium range (62–66 points). Physical functioning (95.40±9.73) and self-

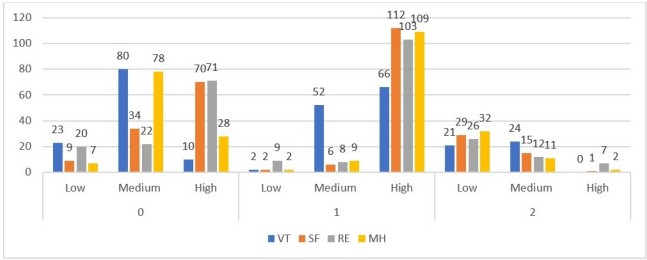

**Fig 6. Distribution of amount of participants by mental health indices in each cluster.**

**Table 5. Presentation of full results put in cluster categories.** Number in brackets indicate the number of participants in a specific group.

| | Subgroup 1*, % (n) | Subgroup 2**, % (n) | Subgroup 3***, % (n) |
|---|---|---|---|
| Students of Military specialization | 27.43 (31) | 34.17 (41) | 42.22 (19) |
| Students of sport specialization | 23.89 (27) | 10.83 (13) | 20.00 (9) |
| Students of internal affair | 27.43 (31) | 25.00 (30) | 17.78 (8) |
| Military service personnel | 21.24 (24) | 30.00 (36) | 20.00 (9) |
| Marital status | | | |
| Divorced/live separate | 2.65 (3) | 1.67 (2) | 0 (0) |
| Single | 87.61 (99) | 94.17 (113) | 82.22 (37) |
| Civil marriage | 2.65 (3) | 0.83 (1) | 0 (0) |
| Marriage | 7.08 (8) | 3.33 (4) | 17.78 (8) |
| Quality of life | | | |
| Physical functioning | | | |
| Low | 1.77 (2) | 0 (0) | 28.89 (13) |
| Medium | 2.65 (3) | 0 (0) | 26.67 (12) |
| High | 95.58 (108) | 100 (120) | 44.44 (20) |
| Physical role functioning | | | |
| Low | 25.66 (29) | 10.83 (13) | 68.89 (31) |
| Medium | 23.89 (27) | 15.83 (19) | 22.22 (10) |
| High | 50.44 (57) | 73.33 (88) | 8.89 (4) |
| Bodily pain | | | |
| Low | 18.58 (21) | 4.17 (5) | 46.67 (21) |
| Medium | 50.44 (57) | 16.67 (20) | 40 (18) |
| High | 30.97 (35) | 79.17 (95) | 13.33 (6) |
| General health | | | |
| Low | 4.42 (5) | 5 (6) | 80 (36) |
| Medium | 84.07 (95) | 20 (24) | 15.56 (7) |
| High | 11.5 (13) | 75 (90) | 4.44 (2) |
| Vitality | | | |
| Low | 20.35 (23) | 1.67 (2) | 46.67 (21) |
| Medium | 70.8 (80) | 43.33 (52) | 53.33 (24) |
| High | 8.85 (10) | 55 (66) | 0 (0) |
| Social Functioning | | | |
| Low | 7.96 (9) | 1.67 (2) | 64.44 (29) |
| Medium | 30.09 (34) | 5 (6) | 33.33 (15) |
| High | 61.95 (70) | 93.33 (112) | 2.22 (1) |
| Emotional role functioning | | | |
| Low | 17.7 (20) | 7.5 (9) | 57.78 (26) |
| Medium | 19.47 (22) | 6.67 (8) | 26.67 (12) |
| High | 62.83 (71) | 85.83 (103) | 15.56 (7) |
| Mental health | | | |
| Low | 6.19 (7) | 1.67 (2) | 71.11 (32) |
| Medium | 69.03 (78) | 7.5 (9) | 24.44 (11) |
| High | 24.78 (28) | 90.83 (109) | 4.44 (2) |

*Subgroup 1 = males with low anxiety and depression, and moderate quality of life;

**Subgroup 2 = males with high quality of life, and low anxiety and depression;

***Subgroup 3 = males with high anxiety and depression, and very low quality of life.

**Table 6. Cluster characteristics with average values (M±SD) of obtained results.**

|  | Subgroup 1* | Subgroup 2** | Subgroup 3*** |
|---|---|---|---|
| Age | 20.29±2.52 | 20.70±2.85 | 20.79±3.5 |
| Military duration | 23.84±47.60 | 26.66±37.29 | 22.38±36.65 |
| Conflict duration | 2.19±4.53 (0–25)[†] | 4.02±8.29 (0–48)[†] | 2.6±5.26 (0–24)[†] |
| Injuries | 0.044 | 0.042 | 0.067 |
| Rehabilitation duration | 0.097 | 0.1 | 0.15 |
| T-anxiety | 44.33±5.93 | 37.37±7.46 | 50.36±6.64 |
| S-anxiety | 42.73±5.93 | 35.35±6.69 | 49.76±7.27 |
| Physical functioning | 95.40±9.73 | 98.42±3.94 | 67.44±28.46 |
| Physical role functioning | 77.00±29.53 | 89.58±19.87 | 46.11±28.68 |
| Bodily pain | 65.64±20.58 | 82.76±15.13 | 47.33±24.05 |
| General health | 65.40±10.10 | 80.52±14.23 | 44.4±13.66 |
| Vitality | 62.84±11.69 | 79.38±12.36 | 51.07±12.95 |
| Social Functioning | 83.30±16.59 | 95.42±9.71 | 50.55±17.66 |
| Emotional role functioning | 78.76±32.44 | 92.50±20.02 | 43.70±34.69 |
| Mental health | 66.41±12.84 | 83.17±10.12 | 47.55±11.97 |

*Subgroup 1 = males with low anxiety and depression, and moderate quality of life;

**Subgroup 2 = males with high quality of life, and low anxiety and depression;

***Subgroup 3 = males with high anxiety and depression, and very low quality of life;

[†]minimum–maximum.

functioning (83.30±16.59 points) were at high levels. Their mean T-anxiety test score was 44.33±5.93 points, and the S-anxiety score was 42.73±5.93 points, which is in the average range of those indicators.

Subgroup 2 was the largest (n = 120) and consisted of people with a mean military duration of 26.66 months and a presence in the conflict zone of 4.02 months. The experience of injuries was the lowest compared to other clusters. This cluster consisted mostly of military services/ universities (64.17%). Almost all indices of quality of life (physical functioning, role-physical, self-functioning, role-emotional) were in very high range (89.58–98.42 points). In general, these participants had the highest quality of life compared to other clusters. Quality of life indices were low according to single scales in only 1.67–7.50% of participants. Of this cluster, 45.0% showed medium or low vitality indices. Their mean T-anxiety test score was 37.37±7.46 points, and the S-anxiety score was 35.35±6.69 points, which is in the low range for those indicators.

Subgroup 3 was the smallest (n = 45) and consisted mainly of students of military (42.22%) or sport (20.00%) specialization or military service personnel (20.00%). Injury experience was at the highest level compared to other clusters. Quality of life scores were at a rather low level, and almost all people in this cluster had low level indices according to role-functioning (68.89% of participants in this group), bodily pain scale (46.67%), general health (80.00%), self-functioning (64.44%), role-emotion (57.78%), and mental health (71.11%). The general quality of life results were not higher than 51 points, except for the physical functioning scale (67.44±28.46 points). Their mean T-anxiety test score was 50.36±6.64 points, and the S-anxiety score was 49.76±7.27 points, which is in the low range for those indicators.

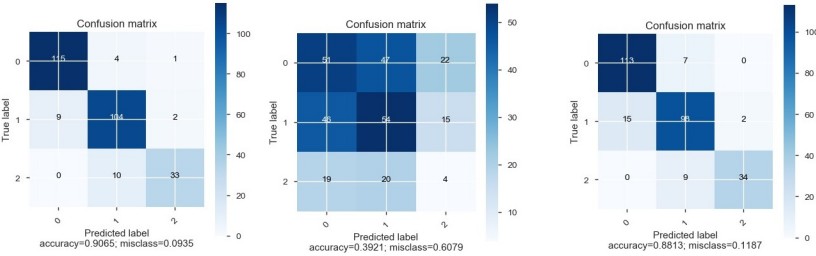

**Fig 7. Confusion matrices.**

## Reliability of model

As a result of classifying and validation, confusion matrixes were obtained that describe the performance of the obtained model. The diagonal element represents the number of points correctly classified, and other elements express mislabelled samples.

The predicted label of accuracy of the model that uses all sampled data (socio-demographic characteristics, quality of life, anxiety results, including data in numeric and categorical form) was 91.01%. Analysis of feature importance does not allow us to highlight individual data sets with a high level of importance that affected the model (relative importance < 0.1), among which the first was high mental health, medium mental health, general health (numerical data), medium general health and mental health (numerical data), vitality (numerical data), S-anxiety (numerical data), and bodily pain (numerical data). The result of the classifier was on a high level for the model that used only quality of life and anxiety features, and the predicted label of accuracy was 86.33% (Fig 7). The features with high importance in this model were general health (0.25) and mental health (0.16) (Fig 8). The predicted accuracy of the model that used socio-demographic data was low (45.32%); among the most important features was age (0.31), military duration (0.31), and conflict duration (0.22) (Fig 9).

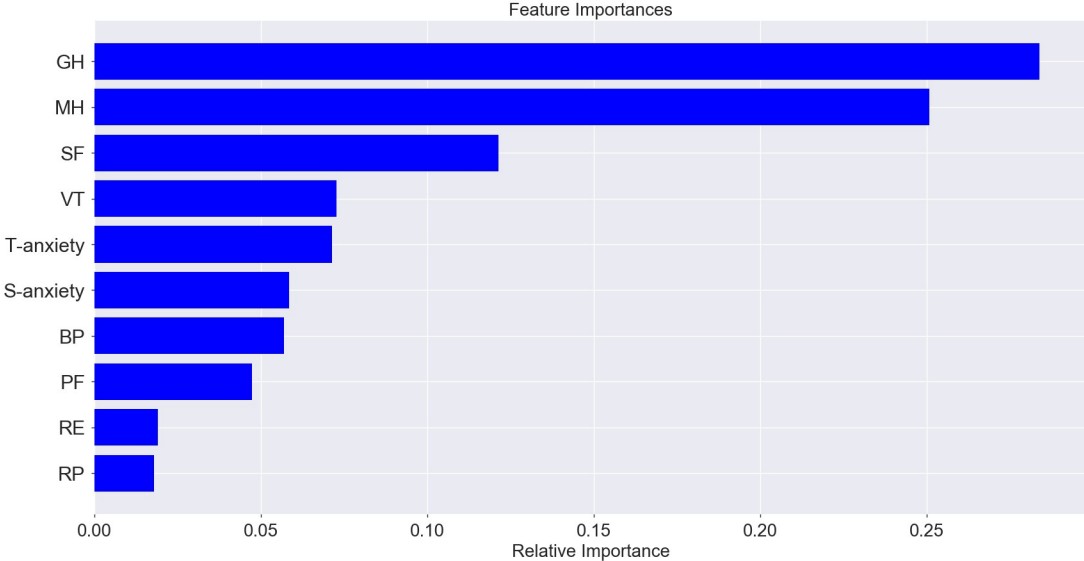

**Fig 8. Proportion of feature importance among quality of life (results of SF-36).**

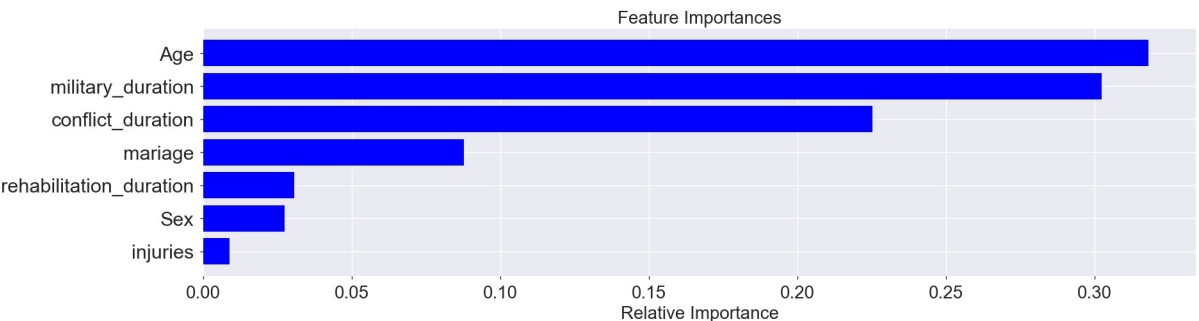

**Fig 9. Proportion of feature importance among socio-demographic indices.**

## Anxiety prediction

The second model with an accuracy of 86.33% was used for the prediction of anxiety level by quality of life data. The three quality of life scales predicted mainly the level of T-anxiety ($R^2 = 0.32$)–mental health (relative importance– 0.28), vitality (0.23), and general health (0.21). S-anxiety was conditioned and could be predicted by mental health (relative importance– 0.58), and only partly by other quality of life components (importance of general health and vitality was less than 0.11); the coefficient of determination for this case was 0.45 (Figs 10 and 11).

## Discussion

In the presented study, we outline that type of perception of quality of life, which plays a significant role in the perception of good well-being, perceived emotion and daily life function [13] in the context of living in a threat of possible engagement in military actions. The applied model of social science computational techniques allowed us to successfully classify participants by their biopsychosocial profiles for 3 subgroups (clusters). What is different in this

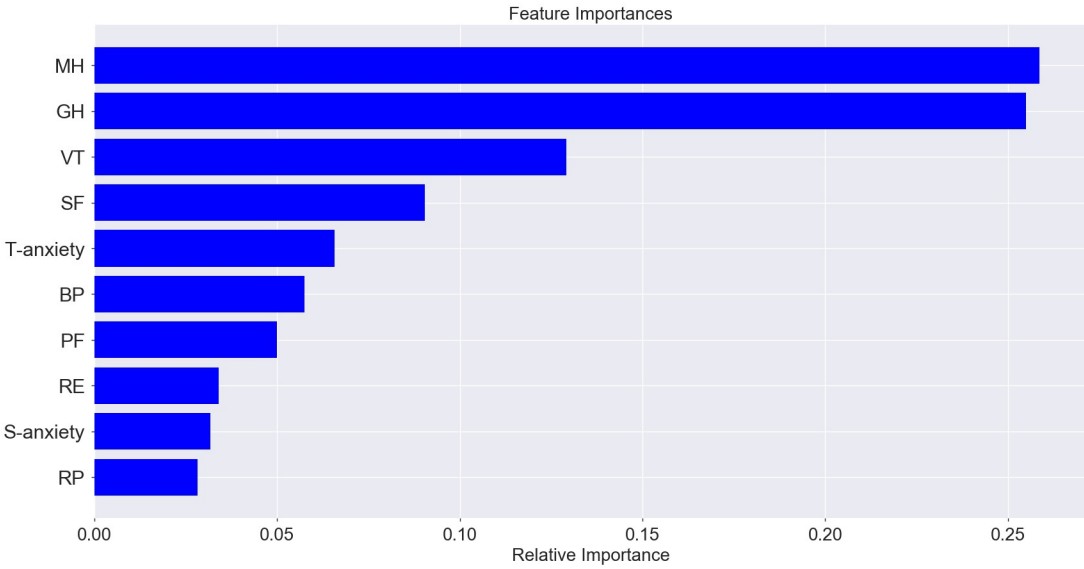

**Fig 10. Feature importance for T-anxiety prediction among quality of life indices.**

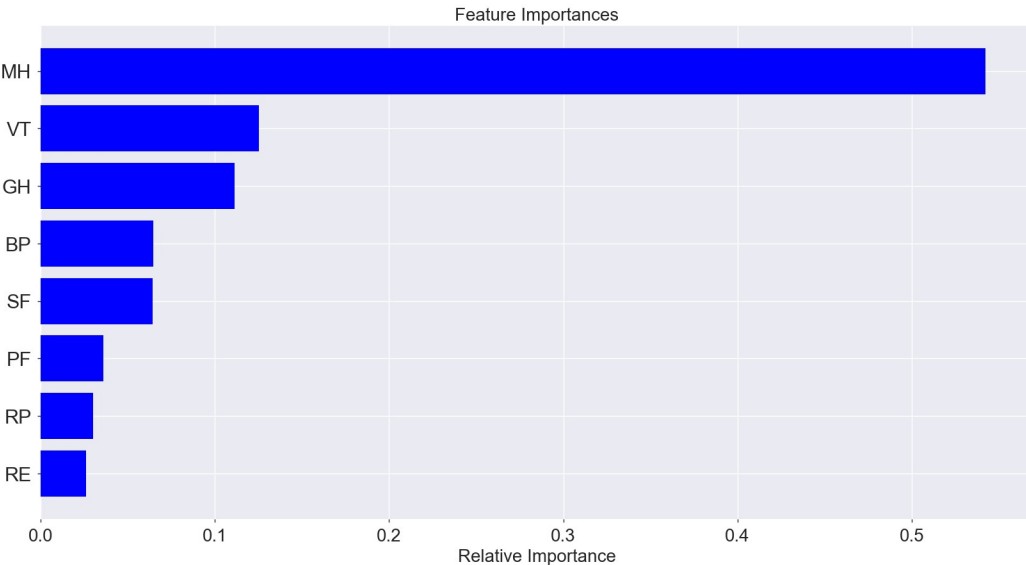

**Fig 11. Feature importance for S-anxiety prediction among quality of life indices.**

presentation of data is that we did not identify profiles for each of the tested groups and describe them, but rather participants from different groups were put together into clusters.

For visualization, we used t-SNE projection of feature space on a 2D plane and concluded that clusters did not have a good geometry. We explained this by the high homogeneity of the group, particularly socio-demographic data (in the study participated young males with a high level of physical performance, and even had particular requirements for physical and mental health). According to previous research, the parameters of quality of life and mental health of these populations were the highest and did not significantly differ.

We did not aim to obtain an algorithm that would give us the highest quality of the predict, and our main goal was to study the dependencies in the data and their impact on the predict of anxiety and depression. We used random forest method that is an ensemble and averages the result of many decision trees. In particular, we tried checking results on cross-validation with and without stratifying based on the target variable, train with and without weights based on the distribution of target variable, train model without changing hyperparameters, and better understand the predictive power of features and internal patterns in data. After such attempts, there was not any significant difference in features importance or accuracy of the model.

Accordingly, it is worth highlighting that anxiety has a role in adapting to the environment, as people who feel anxious have a better ability to detect danger as they are more aware of it. In that case, there is a tendency to receive and label each inconclusive signal as a potential threat [32]. This plays a special role in terms of being anxious due to military operations in the country. When students are aware that they might be conscripted anytime, every possible signal from media could lead to an abnormal fear response. Therefore, there are differences in people's reactions to stressful situations during times of peace and military conflict. These differences are connected with different intensities of anxiousness. This assumption is coherent with the approach of individual differences psychology, which assumes that people are different by specific psychological profiles but are similar to each other in certain aspects, reactions and responses at the same time and that those differences could be measured [15, 16]. This phenomenon reflects a nature of obtained clusters, where are only three general profiles, but each participant has his own unique set of scores, which were not doubled.

The State-Trait Anxiety Inventory (STAI), which was applied in this study, is characterized by both high diagnostic and prognostic validity based on studies on a sample of 1269 young soldiers [33]. The STAI results on both scales allowed us to equally diagnose weakly adopted recruits, as well as diagnose predicting such symptoms in the future. The reliability coefficients for both scales (State and Trait) were similar for both the creators of the STAI [34] and the results of Polish researchers [35, 36]. This allows us to transform results into stens scale in our study and to conclude that in the group of students of military specialization, individual values vary the most in terms of quite constant predispositions towards feeling of anxiety and reacting in the emotional way for environmental stimuli, then interpreting them as threatening. This transformation of results also shows that in every group, there are extremely calm and anxious people at the same time. One of our hypotheses was that level of anxiety will be strictly related to danger level of direct military actions. As students who were conscripted by will be in reserved force are less likely to participate in military operation, we assumed that their anxiety would be lower than actual military personnel or veterans. This turns out to be more complex and individual traits. Proper diagnosis using a presented battery of tests may have a practical impact on further education and professional development of such individuals.

A strong predisposition for feeling anxious may be a source of additional stress in a time of higher threat of being injured. Intense sense of fear, alongside prolonged sensation of pain, could lead to long-term consequences such as overload of the immune system [37]. The obtained results also confirmed dispersity for all studied groups, which was confirmed by cluster analysis, which allowed the identification of three zones of similarities (Fig 2). It indicated a necessity of individualization of the educational process in the institutions connected with preparing soldiers and other participants, where there is strong situational psychic pressure.

It is well known that participation in rescue and military operations can adversely affect physical and mental health. Veterans who participated in hostilities were characterized by increased spending on health care and medical services, the development of post-traumatic stress disorder, increasing the impact of behavioral risk factors [38–40]. However, it should be noted that only participation in specific historical events was the cause of such problems. For example, war veterans in Vietnam and the Persian Gulf were 2–3 times more likely to suffer from physical and mental illness and have a lower quality of life than veterans of the Bosnian conflict [41, 42]. A survey of the British military in Iraq found no differences in physical or mental health or quality of life than those not involved in armed conflict. The risk group can be considered workers (servicemen, law enforcement officers, firefighters) who directly assisted victims of natural disasters, witnessed an emergency or natural disaster, participated in rescue operations, provided assistance to the seriously injured, witnessed the death of children or colleagues, etc [43].

According to world statistics, from 11% to 30% of veterans in different countries face PTSD. There are no reliable statistics for Ukraine, and coverage of veterans' mental health services is insufficient. According to preliminary estimates, the average indicators of psychogenic losses in previous military conflicts were 10–25%. In 2014 among Ukrainian military personnel, they reached almost 80%, and in 30–40% of cases, it may be irreversible losses when psychological problems turn into psychiatric ones. The events taking place in the area of antiterrorist operation are characterized by high intensity, tension, and ephemerality. Activities in such extreme conditions require a person to work at the limit of their capabilities. Currently, there is no specialized institution in Ukraine that would provide relevant services, only the establishment of a department that will specialize in the treatment of severe PTSD, depression, and anxiety disorders in war veterans and their families. Nowadays, there was only an idea to develop a roadmap for care services for veterans and their families and to establish a national Center for Post-Traumatic Syndrome in Ukraine, similar to the one in the United States.

According to a study of the mental state of 189 participants of the anti-terrorist operation, who were in hospitals for war veterans, it was found that volunteers have more acute post-traumatic stress disorder and depression [44]. The volunteers' perceptions of the fighting do not coincide with reality, and they experienced severe stress in this regard. Significant differences between indicators on all scales in those who had combat experience and those who did not have it were not found. First of all, it indicates the unexpected impressions of the participants of the anti-terrorist operation, right on the spot, regardless of their experience.

Differences in anxiety levels have deep neurophysiological backgrounds [32]. There are many genetic or in-born differences, which may cause one person to be immune and another to be more prone to being anxious [35, 45]. Those vulnerabilities for developing a high level of anxiety could be developed or not, depending on environmental factors such as social interactions with others or parenting style in the early age, up to mental or physical trauma that is independent of the age of a person. According to Folkman et al. [46] and Bukova et al. [47], the emotion of fear is connected to the evaluation of difficult situations, which is beyond the control of certain individuals. One of the aspects of this conception is to understand cognitive assessment, emotions and ability to handle them as complex processes characterized by the ability to change in time. Moreover, this concept also indicates that some people could express positive emotions while still remaining under stressful situations. Such positive emotions could accompany those negative feelings as a result of assessing some situations and the expression of adaptability to those challenges. The importance of dependencies such as subjective feelings of general and physical health indicates that physical preparation allows better handling of stress and reduces the consequences of mental stress [48, 49]. Moreover, as military-related groups receive combat training, their preparation was better than others, which may be a cause of the larger number of participants in subgroups 1 and 2. Such phenomena were also confirmed in previous studies [50].

In the context of this conception, we can assume that for those who willingly chose military service or other kind of readiness for engagement in combat, there is a different situation in terms of dealing with anxiety than for those who were put under such stress without any consent (for example, war victims and refugees) [51]. These differences are in the field of interest of health and military psychologists.

In this study, participants were gathered from 4 groups with different levels of proximity and experience with military conflict. The intuitive assumption was that anxiety level will be positively correlated with contact with military service and conflict duration. However, in forming clusters by those socio-demographical dependencies, age was the stronger determinant. However, the accuracy of this prediction of anxiety level was not sufficient (45%). This finding is coherent with the analysis of the Chinese military, where the main determinants of anxiety were also the duration of service [16]. Much higher accuracy in prediction of anxiety was performed by quality of life indices (85.3%), where mental health and general health alongside vitality seem to be far more crucial than socio-demographic factors. Stages of discussion were reversed for purpose in comparison to presented results, as for both forming clusters and predicting anxiety, quality of life, especially mental health, is far more important. Difficult experiences make a risk of lowering mental and somatic health, but this relationship is modified by a vast number of individual factors, as well as social context. There is a correlation between psychological stress and quality of life [52]. Moreover, it seems that those individual profiles affect people more and form stronger connections in the performed cluster analysis than belonging to their social groups. Supposing that belonging to a certain group such as a particular specialization would determine grouping for clusters, there should be 4 groups. However, according to the results of this computation, indices of quality of life were more bonding in terms of assessing anxiety.

From this seemingly complex computation, there are a few quite simple conclusions. First, the group with the lowest level of anxiety also had the highest indices in the quality of life assessment, in addition to the longest duration of military service. This could be explained by some adaptation to those conditions, as facing problems could lower anxiety due to the adaptation process. Representatives from this group were mostly military service personnel, as they have shown the highest values of indices in the SF-36 results. On the other hand, there was a cluster with the same correlation, but low quality of life was associated with the highest anxiety levels. However, this analysis is more complex, as here we must think of this regression as our conclusion, not the engine behind this computation. All factors, despite different affection powers, influence the results. Most likely, without taking into account socio-demographic data, there will be different groups. As this is a neural network, all signals upon entering are put with weights that affect the overall input. These weights change as algorithms learn, but the results depend on every data point at once, which could provide us with more reliable data for discussions, such as the sentence mentioned in this discussion, that psychology is complex.

Moreover, for application value from this study, we could assume the overall anxiety level of someone depends on his quality of life. To lower anxiety levels, the best part will be to consult with psychologists about the individual needs of people instead of trying to solve anxiety problems with one military order. This assumption was confirmed in a previous study [53].

The S-Anxiety and T-Anxiety results, which varied from approximately 35 to 50, were similar to the results of Chinese military personnel, which have indices ranging from approximately 37 to 44 points. Surprisingly, the results presented median values of military students and recruits from the USA showed median values from 54–64 for S-Anxiety and 52–57 for T-Anxiety, while separate groups from this research have obtained median values from approximately 36 up to 50. This means that their quality of life was worse, but political context, ideological motivation and other parts exceeded proper conclusions about this phenomenon [33].

This multilayer analysis, performed by computational techniques instead of a series of individual statistical computations and personal syntheses and interpretations, allowed fresh insight into well-known psychological tests, as putting many seemingly independent data, even data from different questionnaires and test batteries together, could show connections that were previously beyond our grasp.

In terms of military-related samples, machine learning algorithms were used mostly to predict PTSD. Our study did not focus on predicting PTSD chances, rather, quality of life of people nearby warzone, but not necessarily in directs battles. This study's applied values lie in exploring which mental components should be improved and the strong side of the mental status of Ukrainian people. It was shown that their anxiety levels were are lower than participants in other conflicts—soldiers fighting out of their country on missions. Those conditions, defensive character of protecting homeland is unique as first armed conflict in 21th century Europe.

## Conclusions

Machine learning algorithms could be beneficial in the methodology of computational social science, as they could provide deeper insight into collected data. Obtained clusters were composed of people from different types of military-related studies. Therefore, there is no clear relevance between different types of military-related studies or even experiencing military actions and anxiety levels. More factors, mostly subjective feelings about mental conditions, are crucial dependencies in feeling anxiety. Being well prepared physically and mentally (as a port of training/studies) is crucial for the proper handling of stress amount not only for those in actual contact with military conflict but also for those whom such factors are not threatening.

Complex, multifactorial dependencies regarding anxiety levels indicate that rather than general policy towards victims, soldiers, or even civilians, individual approaches and proper identification in particular areas where individuals lack support is crucial for proper prevention towards exceeding the amount of stress.

## Author Contributions

**Conceptualization:** Iuliia Pavlova, Dmytro Zikrach, Dariusz Mosler, Dorota Ortenburger, Jacek Wąsik.

**Data curation:** Iuliia Pavlova, Dmytro Zikrach.

**Formal analysis:** Iuliia Pavlova, Jacek Wąsik.

**Investigation:** Iuliia Pavlova, Dorota Ortenburger, Tomasz Góra, Jacek Wąsik.

**Methodology:** Iuliia Pavlova, Dmytro Zikrach, Dariusz Mosler, Dorota Ortenburger, Jacek Wąsik.

**Project administration:** Iuliia Pavlova, Jacek Wąsik.

**Resources:** Iuliia Pavlova, Dmytro Zikrach.

**Software:** Dmytro Zikrach, Tomasz Góra.

**Supervision:** Iuliia Pavlova, Dmytro Zikrach, Jacek Wąsik.

**Validation:** Iuliia Pavlova, Dmytro Zikrach, Dorota Ortenburger.

**Visualization:** Iuliia Pavlova, Dariusz Mosler, Tomasz Góra.

**Writing – original draft:** Iuliia Pavlova, Dariusz Mosler, Dorota Ortenburger, Tomasz Góra, Jacek Wąsik.

**Writing – review & editing:** Iuliia Pavlova, Dariusz Mosler, Dorota Ortenburger, Jacek Wąsik.

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
