## [Decision Letter · Decision Letter 0]

15 Jul 2020

PONE-D-20-11692

Determinants of anxiety levels among young males in a threat of experiencing military conflict–applying a machine-learning algorithm in a psychosociological study

PLOS ONE

Dear Dr. Mosler,

Thank you for submitting your manuscript to PLOS ONE. After careful consideration, we feel that it has merit but does not fully meet PLOS ONE’s publication criteria as it currently stands. Therefore, we invite you to submit a revised version of the manuscript that addresses the points raised during the review process.

Please address all of the very helpful critical feedback from Reviewer 1 in your revision. You will note that Reviewer 2 was much more critical of the manuscript, so it is important that you consider these broad evaluations of the study when clarifying the purpose, methods and implications of the study.

A rebuttal letter that responds to each point raised by the academic editor and reviewer(s). You should upload this letter as a separate file labeled 'Response to Reviewers'. A marked-up copy of your manuscript that highlights changes made to the original version. You should upload this as a separate file labeled 'Revised Manuscript with Track Changes'.An unmarked version of your revised paper without tracked changes. You should upload this as a separate file labeled 'Manuscript'.

We look forward to receiving your revised manuscript.

Kind regards,

Melita J. Giummarra

Academic Editor

PLOS ONE

Journal Requirements:

2. Please include a separate caption for each figure in your manuscript.

3. Please ensure that you refer to Figure 2 in your text as, if accepted, production will need this reference to link the reader to the figure.

Reviewers' comments:

Reviewer's Responses to Questions

**Comments to the Author**

1. Is the manuscript technically sound, and do the data support the conclusions?

Reviewer #1: Yes

Reviewer #2: No

2. Has the statistical analysis been performed appropriately and rigorously? 

Reviewer #1: Yes

Reviewer #2: I Don't Know

3. Have the authors made all data underlying the findings in their manuscript fully available?

Reviewer #1: Yes

Reviewer #2: Yes

4. Is the manuscript presented in an intelligible fashion and written in standard English?

Reviewer #1: Yes

Reviewer #2: Yes

5. Review Comments to the Author

Reviewer #1: This article presents some interesting results related to the Ukrainian military, while interesting, the article should be reframed to appeal to an international audience. My comments do not detract from the overall quality of the work but seek to improve clarify.

1. The introduction provides an exhaustive explanation of anxiety, and described the role of PTSD in generality, however little attention is focused on the sample of interest, the Ukrainian military. It would be useful for the authors provides a brief summary of the Ukrainian military (any cohort studies?) and perform a comparison to other countries, such as USA, UK and Israel.

2. In addition to the above statement, the authors should provide a brief summary of the current state-of-the-art in the area of machine learning and mental health. It comes as a surprise that classifiers are being employed without any narrative to support the rationale as to why. Some studies I was expecting to see are: [1]–[3]

3. Please clarify why the 36-item Short Form Health Survey wans State-Trait Anxiety Inventory have been used. Have they been validated in military populations?

4. Participants included in the study are conscripts, the authors should provide a brief description on how individuals are selected (and what impact this will have on the results). There are differences between conscript forces and these should be explored.

5. In addition to the above, why are no females included in the sample?

6. Why was only Random Forest used? Did the authors consider the performance of other classifiers? Additional questions:

a. How was the dataset split, was it random or based on participant ID or data ordering?

b. Did the authors seek to balance the classes to ensure proportion of caseness were in each group?

c. Which parameters were specified for the model (n_tree etc)?

7. In terms of clustering, did the authors explore in addition to applying the elbow method the role of cluster compactness using L2 norm? I am curious to see how ‘compact’ each cluster around the centeroid.

8. The authors provide a very detailed and interesting discussion; however, it would be useful to know in more detail how the results of the study compare to the wider Ukrainian military and to provide international comparisons.

9. The manuscript is detailed but lacks citations in numerous places. Please ensure citation are provided for statements and current situations.

10. Typo and grammatical errors throughout, please review further manuscripts carefully.

[1] D. Leightley, V. Williamson, J. Darby, and N. T. Fear, “Identifying probable post-traumatic stress disorder: applying supervised machine learning to data from a UK military cohort,” J. Ment. Heal., vol. 28, no. 1, pp. 34–41, Jan. 2019, doi: 10.1080/09638237.2018.1521946.

[2] K.-I. Karstoft, A. Statnikov, S. B. Andersen, T. Madsen, and I. R. Galatzer-Levy, “Early identification of posttraumatic stress following military deployment: Application of machine learning methods to a prospective study of Danish soldiers,” J. Affect. Disord., vol. 184, pp. 170–175, Sep. 2015, doi: 10.1016/j.jad.2015.05.057.

[3] K. Schultebraucks et al., “Pre-deployment risk factors for PTSD in active-duty personnel deployed to Afghanistan: a machine-learning approach for analyzing multivariate predictors,” Mol. Psychiatry, Jun. 2020, doi: 10.1038/s41380-020-0789-2.

Reviewer #2: The authors do not describe how their sample was selected. Is this meant to be a representative sample? Assuming no, what conclusions do the authors hope to draw about it? The methodology is complex, but with the low N and lack of details about recruitment, I don't see how the authors are able to conclude anything that could contribute to the field. The authors conclusion that machine learning algorithms could be beneficial to social science is neither a new nor compelling message.

6. PLOS authors have the option to publish the peer review history of their article (what does this mean?). If published, this will include your full peer review and any attached files.

Reviewer #1: **Yes: **Dr Daniel Leightley

Reviewer #2: No

---

## [Author Response · Author response to Decision Letter 0]

27 Aug 2020

Dear Reviewers,

It is difficult even to express how grateful we are for reviewing our paper. We want to thank you for your time, a detailed analysis of our article, valuable comments, and suggestions. We tried to make suitable changes accordingly to your comments and recommendations. We would like to describe all these changes, as well as a separate file with all changes in the article, was added.

Reviewer#1

1. The introduction provides an exhaustive explanation of anxiety, and described the role of PTSD in generality, however little attention is focused on the sample of interest, the Ukrainian military. It would be useful for the authors provides a brief summary of the Ukrainian military (any cohort studies?) and perform a comparison to other countries, such as USA, UK and Israel.

Additional part in introduction was added:

“The STAI was used to examining trait and state anxiety in different military population (16-19), and this tool is a recommended method approved by the Supreme Council of Ukraine for diagnostic examination and psychological correction of military personal, especially those individuals who took a direct part in the anti-terrorist operation. However, despite the availability of legal documents, described recommendations and procedures, any analysis or cohort studies of anxiety and depression was conducted on large population groups of Ukrainian military personal, there is no clear data on the prevalence of PTSD among them.”- line 101-107

And:

“Another useful tool is 36-item Short Form Health Survey, which was used for assessing health-related quality of life outcomes, and it is validated for general and military population in different countries (20-22). The questionnaire is widely used to assess the quality of life of the Ukrainian population aged 16–70, in particular for people working in emergency services (23).”- line 119-122

In discussion there is a part about PTSD treatment on the Ukraine:

“Currently, there is no specialized institution in Ukraine that would provide relevant services, only the establishment of a department that will specialize in the treatment of severe PTSD, depression, and anxiety disorders in war veterans and their families. Nowadays, there was only an idea to develop a roadmap for care services for veterans and their families and to establish a national Center for Post-Traumatic Syndrome in Ukraine, similar to the one in the United States.

According to a study of the mental state of 189 participants of the anti-terrorist operation, who were in hospitals for war veterans, it was found that volunteers have more acute post-traumatic stress disorder and depression (44). The volunteers’ perceptions of the fighting do not coincide with reality, and they experienced severe stress in this regard. Significant differences between indicators on all scales in those who had combat experience and those who did not have it were not found. First of all, it indicates the unexpected impressions of the participants of the anti-terrorist operation, right on the spot, regardless of their experience.” – line 488-499

2. In addition to the above statement, the authors should provide a brief summary of the current state-of-the-art in the area of machine learning and mental health. It comes as a surprise that classifiers are being employed without any narrative to support the rationale as to why. Some studies I was expecting to see are: [1]–[3]

Paragraph which contains proposed articles was added:

The idea of using machine-learning algorithms to obtain knowledge about PTSD and anxiety of soldiers is not novel. There were even large cohort studies, including 13 690 participants, in which a supervised machine-learning algorithm was applied (28). The main results of such an experiment conducted by Leightley et al. indicated, that such methods might reduce cost, and helps with earlier detection and prophylaxis (28). Another approach, with feature selection and k-nearest neighbors, was used by Karstof et al., with further confirms the possibility to use a machine learning as a forecast of PTSD symptoms on a group of 561 Danish soldiers (29). Another machine learning classifier random forest, for soldiers that participated in operation in Afghanistan (30). Also, in this case, the sample cannot be considered as big data (n=473). Despite that, machine learning algorithms have proven to be worthy of developing and capable of adjusting to a different group of soldiers participating in various military conflicts. – line 137-148

3. Please clarify why the 36-item Short Form Health Survey wans State-Trait Anxiety Inventory have been used. Have they been validated in military populations?

We gave additionally information about 36-item Short Form Health Survey wans State-Trait Anxiety Inventory as part of the answer to the first question. We hope that these clarifications will be sufficient. 

State-Trait Anxiety Inventory was used to examine trait and state anxiety in different military populations. This tool is a recommended method approved by the Supreme Council of Ukraine for diagnostic examination and psychological correction of military personal, especially those individuals who took a direct part in the anti-terrorist operation. However, despite the availability of legal documents, described recommendations, and procedures. However, despite the availability of legal documents, described recommendations and procedures, any analysis or cohort studies of anxiety and depression was conducted on large population groups of Ukrainian military personal, there is no clear data on the prevalence of PTSD among military personnel. – line 101-107

36-item Short Form Health Survey was used as tool for assessing health-related quality of life outcomes, and it is validated for general and military population in different countries. The questionnaire is widely used to assess the quality of life of the Ukrainian population aged 16-70, in particular for people working in emergency services. – line 119-122

4. Participants included in the study are conscripts, the authors should provide a brief description on how individuals are selected (and what impact this will have on the results). There are differences between conscript forces and these should be explored.

Additional information in the participant section was added:

A sample of Ukrainian males (n=392, M±SD=22.1±5.3) participated in a survey.

Four groups of respondents were involved in the study:

• students of military specialization (n=123, M±SD=21.2±5.8),

• internal affairs specialization (n=101, M±SD=19.1±0.5),

• sport specializations (n=64, M±SD=19.7±1.0),

• military service personnel (n=104, M±SD=25.3±4.7).

The distribution and appointment of conscripts were carried out in proportion to the need and availability of conscription resources, both for the Armed Forces of Ukraine and other military formations. Young people were distributed, taking into account their moral, business, psychological qualities, state of health, physical development, general education, and specialized training, as well as taking into account the need for military reserves. Students from those profiles were most capable for military service, because, in accordance with the curriculum, they were given military-related training such as firearms or proper physical preparation. Although all students in Ukraine had a mandatory conscription process, the directives said that after a minimal amount of preparation time, students from those groups were fastest in terms of being sufficiently trained. -line 173-188

5. In addition to the above, why are no females included in the sample?

Since the beginning of military events in eastern Ukraine, there have been existing restrictions on the appointment of women to military positions. Only in June 2016, following the order of the Ukrainian Ministry of Defense, women were allowed to serve in the combat units of the Armed Forces. In general, in regular units (excluding volunteer battalions), at the time of data collection, 938 military women took part in the anti-terrorist operation. The existence of a gender restriction on women's right to hold military positions is manifested not only during their service but also causes restrictions on the right of women to military education in the relevant specialties. Females are admitted to higher military educational institutions and units for the relevant specialties only according to the list of officer positions for which women can be appointed, approved by the Ministry's order. 

At present, it is clear that restrictions on the rights of women servicemen have been somewhat minimized, but full compliance with the principle of gender equality has not yet been achieved, as some positions in the Armed Forces of Ukraine are still inaccessible to women, especially in special forces and highly mobile landing troops. So at the end of 2019, the combatant status received nearly 10 thousand of women (the total number of such persons is 350 thousand)

According to the General Staff of the Armed Forces of Ukraine, 14% of the total number of servicemen are women. There are 86 women servicemen in command positions of the Armed Forces of Ukraine, including six commanders of medical companies, 78 platoon commanders, and two mortar brigade commanders. 

Given these aspects, the difficulties that women face up during military service, obtaining of military education, and military positions, changes that have taken place in this area in Ukraine, opportunity to form a representative sample only today, in our opinion, a separate study should be devoted to these issues of quality of life and mental health of women that have combatant status.

6. Why was only Random Forest used? Did the authors consider the performance of other classifiers? Additional questions:

a. How was the dataset split, was it random or based on participant ID

or data ordering?

b. Did the authors seek to balance the classes to ensure proportion of

caseness were in each group?

We did not aim to obtain an algorithm that would give us the highest quality of the predict, our main goal was to study the dependencies in the data and their impact on the predict of anxiety and depression. That's why we used Random Forest, because this algorithm is an ensemble and averages the result of many decision trees. In particular, we tried:

1. Checking results on cross validation with and without stratifying based on target variable

2. Train with and without weights based on distribution of target variable

3. Train model without changing hyper parameters, for better understanding predictive power of features and internal patterns in data.

After these experiments, we didn’t see any significant difference in features importance or accuracy of a model. But your recommendations are very interesting and valuable, we will try to build ensemble of different models and compare results with this RF.

c. Which parameters were specified for the model (n_tree etc)?

We used mostly default hyperparameters in scikit-learn implementation (https://scikit-learn.org/stable/modules/generated/sklearn.ensemble.RandomForestClassifier.html), because this gave us better results.

Additionally, all these clarifications were added to the Discussion (lines 416-430)

1. In terms of clustering, did the authors explore in addition to applying the elbow method the role of cluster compactness using L2 norm? I am curious to see how ‘compact’ each cluster around the centeroid.

For visualization, we used t-SNE projection of feature space on 2D plane, and we were able to make a conclusion that clusters did not have good geometry. Based on this, we skipped the next steps regarding checking the “geometry” of clusters, for example, compactness, distribution inside, and distance between clusters. 

We explain this by the high homogeneity of the group, in particular socio-demographic data (young male, high level of physical performance, special requirements to the level of physical and mental health). According to our previous research, the parameters of quality of life and mental health of these populations were the highest and did not differ – lines 416-421

2. The authors provide a very detailed and interesting discussion; however, it would be useful to know in more detail how the results of the study compare to the wider Ukrainian military and to provide international comparisons.

In order to expand the discussion and clarify some points, we have added additional paragraphs to the discussion.

“It is well known that participation in rescue and military operations can adversely affect physical and mental health. Veterans who participated in hostilities were characterized by increased spending on health care and medical services, the development of post-traumatic stress disorder, increasing the impact of behavioral risk factors (38-40). However, it should be noted that only participation in specific historical events was the cause of such problems. For example, war veterans in Vietnam and the Persian Gulf were 2–3 times more likely to suffer from physical and mental illness and have a lower quality of life than veterans of the Bosnian conflict (41-42). A survey of the British military in Iraq found no differences in physical or mental health or quality of life than those not involved in armed conflict. The risk group can be considered workers (servicemen, law enforcement officers, firefighters) who directly assisted victims of natural disasters, witnessed an emergency or natural disaster, participated in rescue operations, provided assistance to the seriously injured, witnessed the death of children or colleagues, etc (43).

According to world statistics, from 11% to 30% of veterans in different countries face PTSD. There are no reliable statistics for Ukraine, and coverage of veterans' mental health services is insufficient. According to preliminary estimates, the average indicators of psychogenic losses in previous military conflicts were 10–25%. In 2014 among Ukrainian military personnel, they reached almost 80%, and in 30–40% of cases, it may be irreversible losses when psychological problems turn into psychiatric ones. The events taking place in the area of anti-terrorist operation are characterized by high intensity, tension, and ephemerality. Activities in such extreme conditions require a person to work at the limit of their capabilities. Currently, there is no specialized institution in Ukraine that would provide relevant services, only the establishment of a department that will specialize in the treatment of severe PTSD, depression, and anxiety disorders in war veterans and their families. Nowadays, there was only an idea to develop a roadmap for care services for veterans and their families and to establish a national Center for Post-Traumatic Syndrome in Ukraine, similar to the one in the United States.”

“According to a study of the mental state of 189 participants of the anti-terrorist operation, who were in hospitals for war veterans, it was found that volunteers have more acute post-traumatic stress disorder and depression (44). The volunteers’ perceptions of the fighting do not coincide with reality, and they experienced severe stress in this regard. Significant differences between indicators on all scales in those who had combat experience and those who did not have it were not found. First of all, it indicates the unexpected impressions of the participants of the anti-terrorist operation, right on the spot, regardless of their experience.” -lines 467 - 499

9. The manuscript is detailed but lacks citations in numerous places. Please ensure citation are provided for statements and current situations.

We added proper citation in numerous places, which results in extending number of citations from 30 to 53.

10. Typo and grammatical errors throughout, please review further manuscripts carefully. 

We reviewed our work and corrected all found errors.

 

Reviewer #2: 

The authors do not describe how their sample was selected. Is this meant to be a representative sample? Assuming no, what conclusions do the authors hope to draw about it? The methodology is complex, but with the low N and lack of details about recruitment, I don't see how the authors are able to conclude anything that could contribute to the field. The authors conclusion that machine learning algorithms could be beneficial to social science is neither a new nor compelling message. 

We tried to clarify the information about sample and added additional information in methods. 

“A sample of Ukrainian males (n=392, M±SD=22.1±5.3) participated in a survey.

Four groups of respondents were involved in the study:

• students of military specialization (n=123, M±SD=21.2±5.8),

• internal affairs specialization (n=101, M±SD=19.1±0.5),

• sport specializations (n=64, M±SD=19.7±1.0),

• military service personnel (n=104, M±SD=25.3±4.7).

The distribution and appointment of conscripts were carried out in proportion to the need and availability of conscription resources, both for the Armed Forces of Ukraine and other military formations. Young people were distributed, taking into account their moral, business, psychological qualities, state of health, physical development, general education, and specialized training, as well as taking into account the need for military reserves. Students from those profiles were most capable for military service, because, in accordance with the curriculum, they were given military-related training such as firearms or proper physical preparation. Although all students in Ukraine had a mandatory conscription process, the directives said that after a minimal amount of preparation time, students from those groups were fastest in terms of being sufficiently trained. – lines 173 - 188

Universities offers reserve officer training for two years. An alternative to that is one-year regular service as conscript troops. Maximum conscription age is 27 years.

Students in our sample were conscripted for ground type troops such as mechanized infantry, artillery and engineering troops. Total amount of Ukrainian ground forces are 145 000 soldiers. According to good statistical practice, Cochran’s Sample Size Formula was applied. With 5% of margin of error and confidence level of 95%, required sample size was 375 participants, therefore our sample was representative (31).” – lines 191 - 195

Also additional section in the discussion was added:

“In terms of military-related samples, machine learning algorithms were used mostly to predict PTSD. Our study did not focus on predicting PTSD chances, rather, quality of life of people nearby warzone, but not necessarily in directs battles. This study's applied values lie in exploring which mental components should be improved and the strong side of the mental status of Ukrainian people. It was shown that their anxiety levels were are lower than participants in other conflicts – soldiers fighting out of their country on missions. Those conditions, defensive character of protecting homeland is unique as first armed conflict in 21th century Europe.” – lines 575 - 581

Besides provided explanations and formed conclusions, we believe that sole contribution to the psychological state of Ukrainians during this conflict could contribute for other researchers interested in anxiety and PTSD topics, as this group is another comparable set of data. Even if suggests that machine-learning techniques were not novel or were not the issues of high interest, the originality of this paper also lies in possibility of providing those data about rather unique sample in a context of English-written papers.

---

## [Decision Letter · Decision Letter 1]

14 Sep 2020

Determinants of anxiety levels among young males in a threat of experiencing military conflict–applying a machine-learning algorithm in a psychosociological study

PONE-D-20-11692R1

Dear Dr. Mosler,

We’re pleased to inform you that your manuscript has been judged scientifically suitable for publication and will be formally accepted for publication once it meets all outstanding technical requirements.

Kind regards,

Ethan Moitra

Academic Editor

PLOS ONE

Additional Editor Comments (optional):

Reviewers' comments:

Reviewer's Responses to Questions

**Comments to the Author**

1. If the authors have adequately addressed your comments raised in a previous round of review and you feel that this manuscript is now acceptable for publication, you may indicate that here to bypass the “Comments to the Author” section, enter your conflict of interest statement in the “Confidential to Editor” section, and submit your "Accept" recommendation.

Reviewer #1: All comments have been addressed

2. Is the manuscript technically sound, and do the data support the conclusions?

Reviewer #1: Yes

3. Has the statistical analysis been performed appropriately and rigorously? 

Reviewer #1: Yes

4. Have the authors made all data underlying the findings in their manuscript fully available?

Reviewer #1: No

5. Is the manuscript presented in an intelligible fashion and written in standard English?

Reviewer #1: Yes

6. Review Comments to the Author

Reviewer #1: (No Response)

7. PLOS authors have the option to publish the peer review history of their article (what does this mean?). If published, this will include your full peer review and any attached files.

Reviewer #1: No

---

## [Editor Report · Acceptance letter]

23 Sep 2020

PONE-D-20-11692R1 

Determinants of anxiety levels among young males in a threat of experiencing military conflict–applying a machine-learning algorithm in a psychosociological study 

Dear Dr. Mosler:

I'm pleased to inform you that your manuscript has been deemed suitable for publication in PLOS ONE. Congratulations! Your manuscript is now with our production department. 

Kind regards, 

on behalf of

Dr. Ethan Moitra 

Academic Editor

PLOS ONE